# What Are the Reasons for the Different COVID-19 Situations in Different Cities of China? A Study from the Perspective of Population Migration

**DOI:** 10.3390/ijerph18063255

**Published:** 2021-03-21

**Authors:** Yanting Zheng, Jinyuan Huang, Qiuyue Yin

**Affiliations:** 1School of Economics and Resource Management, Beijing Normal University, Beijing 100875, China; jinyuan1996@yeah.net (J.H.); 201921410013@mail.bnu.edu.cn (Q.Y.); 2Beijing Key Lab of Study on SCI-TECH Strategy for Urban Green Development, Beijing 100875, China

**Keywords:** COVID-19, epidemic spreading, epidemic prevention and control, population migration

## Abstract

Understanding the reasons for the differences in the spread of COVID-19 in different cities of China is important for future epidemic prevention and control. This study analyzed this issue from the perspective of population migration from Wuhan (the epicenter of the COVID-19 outbreak in China). It reveals that population outflow from Wuhan to other cities in Hubei Province (the province where Wuhan is located) and metropolises and provincial capitals outside of Hubei province exceeded those to other cities. This is broadly consistent with the distribution of confirmed COVID-19 cases. Additionally, model analysis revealed that population outflow from Wuhan was the key factor that determined the COVID-19 situations. The spread of COVID-19 was positively correlated with GDP per capita and resident population and negatively correlated with the distance from Wuhan and the number of hospital beds, while population density was not a strong influential factor. Additionally, the demographic characteristics of population migration from Wuhan also affected the virus transmission. Particularly, businesspeople (who tend to have a high frequency of social activities) were more likely to spread COVID-19. This study indicated that specific measures to control population outflow from the epicenter at the early stage of the epidemic were of great significance.

## 1. Introduction

In December 2019, an outbreak of COVID-19 occurred in Wuhan, China. By 30 January 2020, COVID-19 had rapidly spread to 31 provinces (or autonomous regions or municipalities directly under the control of the central government) of China. As of March 2021, COVID-19 has been observed in 214 countries and has resulted in huge losses of human life and damage to the global economy. The number of deaths from COVID-19 has surged exponentially, with more than 1.8 million deaths worldwide by the end of 2020.

Due to the human-to-human transmission capability of the severe acute respiratory syndrome coronavirus 2 (SARS-CoV-2)—the virus that causes the disease COVID-19 (Chan et al., 2020) [1]—contact with people, especially those from Wuhan (the outbreak epicenter), was the main channel of the spread of COVID-19 in China. However, as this article has noted, the COVID-19 situation is not directly determined by population migration from Wuhan. For instance, as of 3 March 2020, the cumulative number of confirmed COVID-19 cases in Xinyang was 50% of that in Chongqing, although the population migration from Wuhan to Xinyang during the Spring Festival travel rush was larger than that to Chongqing. This leads to the question: What are the reasons for the differences in the COVID-19 situations in different cities of China?

On 23 January 2020, two weeks after the start of the Spring Festival travel period (10 January 2020), the Wuhan government announced travel restrictions to curb the spread of COVID-19. During these two weeks, about five million people left Wuhan due to the Spring Festival and the COVID-19 outbreak [2]. According to the estimations of Du et al. (2020) [3], 131 cities of China had a probability in excess of 50% that COVID-19 patients had been imported from Wuhan to the city before the announcement of Wuhan’s travel restrictions, which may have triggered the COVID-19 outbreaks in these cities. Therefore, understanding the distribution and characteristics of population migration from Wuhan in 10–23 January 2020, as well as their relationships with the spread of COVID-19, is highly important to clarify the differences in the COVID-19 situations in different Chinese cities and identify effective measures for the prevention and control of the disease.

Since the outbreak of COVID-19, its spread has been intensively investigated, including the transmission dynamics (Wu et al., 2020a) [4], forecasting using the SEIR model (Wu et al., 2020b; Zhan et al., 2020) [5,6], and the performance of epidemic prevention measures (Chinazzi et al., 2020; Zhang et al., 2020) [7,8]. However, most of these studies focused on the life-science part of this issue, while the impact of social factors and urban fundamentals on the transmission of COVID-19 has been neglected. For this reason, Liu (2020) established a linear regression model to explore the effects of social, economic, and geographical factors on the spread of COVID-19 [9]. Therein, the geographical distance from Wuhan was used as the proxy variable of population migration. The study of Liu (2020) revealed that the geographical distance from Wuhan is a very strong influencing factor and is negatively correlated with the spread of COVID-19 [9]. However, some scholars have argued that there is a more complicated relationship between the spread of major public health emergencies and the movement of people between cities. It is not necessarily the case that the closer a city is to the epicenter of such events, the greater the negative effects it will suffer [10]. As demonstrated in the present study, as well as the distance from Wuhan, the number of confirmed COVID-19 cases in the target city is also highly related to the population that migrated from Wuhan.

Some studies have focused on the connection between migration and the spread of the COVID-19 epidemic. Sirkeci and Yucesahin (2020) concluded that if countries were aware of the importance of migration at the early stage of the epidemic, monitoring immigrant stock data and travel volume data based on human mobility corridors, they could have taken advance measures to prevent the domestic spread of the epidemic [11]. Additionally, Fan et al. (2020a) used data from the 2017 China migrant population survey to construct a residence–birthplace matrix to analyze and forecast the spread of the epidemic [12]. Moreover, Jia et al. (2020) and Fan et al. (2020b) also found that the distribution of population outflow from Wuhan can accurately predict the relative frequency and geographical distribution of COVID-19 infections [13,14]. Kraemer et al. (2020) further used real-time mobility data from Wuhan and data from 554 confirmed cases with travel history in Wuhan to elucidate the role of migration in urban epidemic transmission [15]. They found that mobility out of Wuhan was the main driver of COVID-19 spread before Wuhan’s travel restrictions. However, these studies did not analyze the differences in the epidemic situation of different cities from the perspective of multiple factors, nor did they pay attention to the differences in the characteristics of the population migrating from Wuhan and the differences in the epidemic situation of each city that might be caused by them. Therefore, in the present study, by combining social and economic factors, we tried to clarify the reasons for the differences in the COVID-19 situations in different cities in China, especially those outside Hubei Province, from the perspective of the size and characteristics of population outflow from Wuhan.

Currently, the spread of COVID-19 in China has been controlled, while that in other countries is still rapidly developing. Therefore, a review of the spread of COVID-19 in China is essential to provide references in social and economic terms for the prevention and control of COVID-19 and other major public health events in the future in other countries.

The remainder of this article is structured as follows. Section 2 presents the research methods and data sources. Section 3 presents the results and analysis, in which the distributions of confirmed COVID-19 cases are thoroughly discussed and population migration from Wuhan is estimated. Based on this, a regression model was established to investigate the differences in the spread of COVID-19 in different Chinese cities. Furthermore, the effects of demographic characteristics on the spread of COVID-19 were investigated. Finally, Section 4 presents the conclusions.

## 2. Materials and Methods

### 2.1. Data

Three types of data were collected in this study.

(1) Data of confirmed COVID-19 cases. These data were collected from the websites of health commissions of different cities. The numbers of confirmed COVID-19 cases in different cities of China until 23 January 2020, 8 February 2020, 17 February 2020, and 3 March 2020 were collected, respectively.

(2) Big data of migration and distance. Big data of Spring Festival migration were obtained from the Baidu map insight platform [16]. This dataset includes migration scale index data that are viable for transverse comparison, and the outflow/inflow population of a city to/from another city as a percentage of the overall outflow/inflow population of this city. Zhan et al. (2020) and Kraemer et al. (2020) also used the migration scale index to help analyze and predict the spread of the epidemic [6,15]. Migration scale index data from outflow population migration from Wuhan between 10 January and 3 March 2020 and the same period in the lunar calendar in 2019 were collected from the Baidu map insight platform. Additionally, the outflow population migration from Wuhan to different cities as a percentage of the overall outflow population migration from Wuhan between 10 January and 26 January 2020 was also obtained from the same platform. Furthermore, the road distances from different cities to Wuhan were obtained from the Baidu Maps [17]. Migration data were obtained for 139 prefecture-level cities and were used in the regression. Figure 1 shows the distribution of the studied cities and marks the key cities that were focused on in the subsequent analysis.

(3) Population, economic, healthcare, and urban construction data. Resident populations of different cities were obtained from official 2019 statistical yearbooks published by provincial statistics bureaus; GDP per capita and the number of hospital beds were obtained from the China City Statistical Yearbook 2018; population density, road area per capita, public recreational green space per capita, transported garbage, and discharged wastewater were obtained from the China City Statistical Yearbook 2017.

### 2.2. Methods

The geographical distribution of confirmed COVID-19 cases in China was firstly analyzed. To illustrate the evolutionary history of COVID-19, the numbers of confirmed COVID-19 cases on four key dates were analyzed. The first date is 23 January 2020, which is the date of the announcement of travel restrictions by the Wuhan government. On this date, public transportation, subways, ferries, and long-distance passenger transportation in Wuhan were suspended and airports and railway stations in Wuhan were shut down. The second date is 8 February 2020, which is 15 days after the start of the Spring Festival. On this date, people who had migrated from Wuhan to various other cities in China before the Spring Festival were discharged after the incubation period of COVID-19 (around 14 days), which means that, by this day, COVID-19 patients who had migrated from Wuhan to the city had largely been detected and documented. Therefore, reporting the number of confirmed COVID-19 cases in each city on this date can provide a more comprehensive display of the infected people that had moved from Wuhan. Considering that the COVID-19 patients exported from Wuhan would have infected more people in the target city, we continued to collect the numbers of confirmed COVID-19 cases on the following key dates, namely, the third and fourth dates. The third date is 17 February 2020, which is the date when the number of existing confirmed COVID-19 cases in China reached its peak (58,097 cases). The fourth date is 3 March 2020, which is the date when the first confirmed imported case of COVID-19 outside of China was observed. Data after 3 March 2020 were excluded to avoid the influence of imported cases.

Secondly, the geographical distribution characteristics of population migration from Wuhan during the COVID-19 outbreak were analyzed based on estimations of population migration from Wuhan. According to the trend of migration from Wuhan during 10–26 January 2020, the daily migration scale index of Wuhan exceeded 0.66. Subsequently, between 27 January and 3 March 2020, the daily migration scale index of Wuhan remained below 0.5, suggesting that population migration from Wuhan during the Spring Festival was concentrated in 10–26 January 2020. Additionally, reports indicate that, on 26 January 2020, about five million people left Wuhan due to the Spring Festival and the COVID-19 outbreak [2]. Later, between 27 January and 3 March 2020, population migration from Wuhan decreased by 98% compared with the same period in the 2019 lunar calendar (Li et al., 2020) [18]. Therefore, we assumed that five million people left Wuhan before January 26, and selected the population flow from 10 to 26 January for key analysis. Population migration from Wuhan to different cities during this period was estimated. First, the daily population migration from Wuhan to each target city as a percentage of the overall population migration from Wuhan was multiplied by the daily migration scale index of Wuhan. Then, the results were summarized to calculate the total population migration from Wuhan to each target city as a percentage of the overall population migration from Wuhan during 10–26 January 2020. Finally, the total population migration from Wuhan to the target city during this period was estimated by multiplying the corresponding percentage by five million, which is the overall population that migrated from Wuhan during this period. Given the data availability, we estimated the data for 139 prefecture-level cities.

Thirdly, a linear regression model was established with the number of confirmed COVID-19 cases in the target city as the dependent variable. The effects of population, economy, urban construction, and healthcare conditions on the dependent variable were investigated, especially the effects of population migration from Wuhan. Considering the differences between cities in Hubei Province and those outside the province, the full sample and a subsample without cities in Hubei Province were analyzed separately. The regression equation is as follows:(1)$${\mathrm{disease}}_{i}=\beta_{0}+\beta_{1}{\mathrm{lnx}}_{1i}+\beta_{2}{\mathrm{lnx}}_{2i}+\ldots +\beta_{k}{\mathrm{lnx}}_{\mathrm{ki}}+\varepsilon_{i}$$
where ${\mathrm{disease}}_{i}$ refers to the accumulated number of confirmed COVID-19 cases in the target city, $x_{\mathrm{ki}}$ refers to the factors affecting the number of confirmed COVID-19 cases in the target city, $\varepsilon_{i}$ refers to the random error, and i refers to the target city. Normal transformation is performed on all the dependent variables in the model to ensure a normal error distribution. Additionally, to eliminate possible heteroscedasticity, the heteroscedasticity-robust standard error was used in the estimation process, and the heteroscedasticity-robust t statistic was used to test the significance of the coefficients. The population migration from Wuhan was selected as the target independent variable. Additionally, a series of control variables were considered.

First, since road distance may be an important factor affecting the transmission of the virus, we included the distance between each city and Wuhan in the regression model (Liu, 2020) [9]. Specifically, the shortest distance between the municipal governments of each city and Wuhan along a road network was obtained. We chose the locations of city governments to calculate the distance, since the city government is usually located in the center of dense urban population and economic activities, and therefore, it is the major traffic node for the spread of the epidemic. Second, as provinces with higher economic development tend to attract more population inflow, which may lead to a greater risk of epidemic transmission (Li and He, 2020) [19], we used GDP per capita and resident population as variables, respectively. Third, since the level of healthcare in each city may directly affect the spread of the epidemic, the number of hospital beds—which can be used to indicate the level of healthcare—was used as a variable (Li and He, 2020) [19]. Fourth, because densely populated areas may accelerate the spread of COVID-19, we used population density as a variable (Liu et al., 2020) [20]. Fifth, following Liu (2020), we controlled for four urban construction variables, namely road area per capita, domestic garbage per capita, discharged wastewater per capita, and public recreational green space per capita [9]. Finally, compared with other cities, provincial capitals may have a closer connection with Wuhan, with more tourism and business flow between them, and are more vulnerable to influences (Shi and Liu, 2020) [21]; therefore, we added dummy variables for whether the city is a provincial capital. The definitions of these variables are listed in Table 1. We conducted a Jarque-Bera diagnosis and the *p*-values of the Jarque-Bera test for all transformed dependent variables were greater than 0.66, indicating that the dependent variables obey a normal distribution. Table 2 reports the correlation coefficients between the explanatory variables. It can be seen that the maximum correlation coefficient among the explanatory variables is 0.591. Furthermore, the variance inflation factor (VIF) of each explanatory variable was calculated, and it was found that all the VIF values were less than 3, indicating that there was no serious multicollinearity between the explanatory variables.

Finally, the significant differences in the spread of COVID-19 in different cities were further investigated from the perspective of the demographic characteristics of population migration from Wuhan.

## 3. Results and Discussion

### 3.1. Geographical Distribution of Confirmed COVID-19 Cases

Figure 2 shows the accumulated numbers of confirmed COVID-19 cases in different cities of China on 23 January, 8 February, 17 February, and 3 March 2020, respectively. It was observed that the geographical distribution of confirmed COVID-19 cases exhibited two features.

First, confirmed COVID-19 cases apparently diffused to neighboring regions. On 23 January 2020, the confirmed COVID-19 cases were largely concentrated in Wuhan (495), with less than 30 confirmed COVID-19 cases in other cities. On 8 February 2020, the numbers of confirmed COVID-19 cases exceeded 400 in 13 cities, 11 of which were in Hubei, mainly near Wuhan. The situations on 17 February and 03 March followed similar evolution trends, with confirmed cases spreading in the vicinity of Wuhan.

Second, the numbers of confirmed COVID-19 cases in metropolises and provincial capitals were larger than those in other cities outside of Hubei Province. On 8 February 2020, the 13 cities with over 400 confirmed COVID-19 cases included 11 cities in Hubei Province, as well as Chongqing and Wenzhou. On 17 February, the accumulated numbers of confirmed COVID-19 cases in the metropolises of Shenzhen, Beijing, Guangzhou, and Shanghai exceeded 300. It was observed that, on this date, in 24 out of 26 Chinese provinces and autonomous regions (i.e., all except Hubei and Taiwan), the number of confirmed COVID-19 cases in the provincial capital was among the top three of all cities in that province or autonomous region. Hence, it can be concluded that provincial capitals are more susceptible to the spread of COVID-19. Therefore, in the following model analysis, ‘provincial capital or not’ was employed as one of the control variables.

### 3.2. Estimation and Analysis of Population Migration from Wuhan

Figure 3 depicts the population migration from Wuhan during the Spring Festival travel period. As shown in the figure, a peak of population migration from Wuhan was observed during 21–23 January, and the migration scale index was maximum (11.84) on 22 January. In the same period in the lunar calendar of 2019, the index was 9.6. According to statistics released by the Weibo account Wuhan Railway, on 22 January 2020, 299,600 passengers travelled from three railway stations in Wuhan [22]. By considering the uncounted people who left by highway and waterway transportation, the total population outflow from Wuhan is likely to have been even higher. This may be attributed to an announcement made by Professor Zhong Nanshan on 20 January 2020, in which he confirmed human-to-human transmission of SARS-CoV-2 [23], which is likely to have prompted a large number of people to outflow from Wuhan. After 23 January 2020, population migration from Wuhan dropped drastically, and after 26 January 2020, population migration from Wuhan remained at a relatively low level for the next two months (migration scale index <0.5). This suggests that the travel restrictions imposed by the Wuhan government effectively reduced population migration from Wuhan. Therefore, population migration from Wuhan before 26 January 2020 and the destinations of this migration were chosen as focuses of this study.

The population migration from Wuhan to different cities during 10–26 January 2020 was thoroughly investigated by estimating the number of people outflowing from Wuhan to the target city during this period. The population outflow from Wuhan during this period exhibited several geographical distribution characteristics.

In total, 76% of the population outflow from Wuhan was to other cities in Hubei Province, which is basically consistent with the outflow estimated by Fan et al. (2020), using data from the 2013–2018 China Migrants Dynamic Survey [14]. Of the cities in Hubei, Xiaogan received the largest population migration from Wuhan (750,000), followed by Huanggang (710,000) (see Figure 4). This can explain the fact that Xiaogan and Huanggang became the two cities with the largest number of confirmed COVID-19 cases in China apart from Wuhan. Additionally, five other cities in Hubei—namely Jingzhou, Xianning, E’zhou, Xiangyang, and Huangshi—received a population migration from Wuhan of over 200,000. Compared to inter-provincial population migration, inter-city population migration within one province tends to be significantly more frequent. Moreover, Wuhan’s role as a provincial capital increases its population migration to and from other cities in Hubei. Therefore, in the following analysis, we also examine the effects of population migration on the spread of COVID-19 by focusing on the samples outside of Hubei Province.

Aside from other cities in Hubei, cities in Henan Province received the largest population migration from Wuhan (290,000), followed by cities in Hunan Province (180,000) (see Figure 5). This can be attributed to the fact that both Henan and Hunan border Hubei, which facilitates inter-provincial communication and population migration and simultaneously puts more pressure on the prevention and control of the spread of COVID-19. Additionally, populations of approximately 100,000 migrated from Wuhan to cities in the provinces of Anhui and Jiangxi, respectively, which also border Hubei. Moreover, a population of approximately 40,000 migrated from Wuhan to the provinces of Guangdong, Jiangsu, and Zhejiang, and the municipality of Beijing, respectively, which is larger than other provinces that do not border Hubei.

Besides cities in provinces bordering Hubei, cities in provinces which do not border Hubei with over 11,000 migrants from Wuhan mainly included Beijing, Shanghai, Shenzhen, and Guangzhou, which are metropolises or provincial capitals with populations over 10 million (see Figure 6); specifically, population migrations from Wuhan to Beijing, Shanghai, Shenzhen, and Guangzhou were 47,000, 36,000, 27,000, and 27,000, respectively. Meanwhile, population migrations from Wuhan to provincial capitals such as Chengdu, Nanjing, Hangzhou, and Kunming all exceeded 11,000. These metropolises and provincial capitals are characterized by high population density, frequent economic activities, and large numbers of migrants from Wuhan to these cities are more likely to subsequently migrate to other cities, resulting in the further spread of COVID-19.

### 3.3. Model Results and Analysis

What is the specific relationship between the number of confirmed COVID-19 cases in the target city and population migration from Wuhan to the target city? Moreover, are there other social and economic factors affecting the spread of COVID-19? These questions have not been fully answered in previous studies. Based on the distributions of confirmed COVID-19 cases and population migrations, a linear regression model was established to illustrate the effects of these factors on the spread of COVID-19. Specifically, the accumulated numbers of confirmed COVID-19 cases until 23 January 2020, 8 February 2020, 17 February 2020, and 3 March 2020, respectively, were collected and used as the dependent variable. The full sample and the subsample without cities in Hubei Province were analyzed separately. Table 3 shows the results of the proposed model.

First, the results showed that the goodness of fit of the model with the accumulated number of confirmed COVID-19 cases until 23 January 2020, as the dependent variable was lower than those of other models (columns 1 and 5 in Table 3). This may be due to the fact that the nucleic acid-based detection of COVID-19 had not been fully popularized at this date, meaning that the number of confirmed COVID-19 cases reported at the early stage of the epidemic was not fully consistent with the actual number of confirmed cases. Li et al. (2020) claimed that 86% of COVID-19 patients were not confirmed before 23 January 2020 [18]. Additionally, the actual situation of the spread of COVID-19 had not been fully revealed on 23 January 2020, as SARS-CoV-2 has an incubation period of around 14 days.

Second, the coefficient of population migration from Wuhan was significantly positive at 1%, indicating significant effects of population migration from Wuhan on the spread of COVID-19 in the target cities. Therefore, to a statistically significant degree, population migration, especially from the outbreak center, accelerated the spread of COVID-19. For this reason, it can be concluded that the travel restrictions imposed by the Wuhan government effectively cut off the transmission route of COVID-19 by reducing the migration of SARS-CoV-2 carriers to other cities, thus exerting a significant positive effect on the spread of COVID-19 in other cities. Additionally, the distance from Wuhan to the target city was significantly positive, indicating that the number of confirmed COVID-19 cases in the target city increased as its distance from Wuhan decreased. This is consistent with the study of Liu (2020) [9]. Nevertheless, with the population migration from Wuhan as the independent variable, the distance from Wuhan to the target city was not as important as was found in this previous study.

Finally, other variables reflected other trends of the spread of COVID-19. The number of confirmed COVID-19 cases in the target city had a significant positive correlation with its GDP per capita and resident population, indicating that cities with a more developed economy and larger population will have more confirmed COVID-19 cases. For the subsample without Hubei Province, the coefficient of the number of hospital beds in the target city was significantly negative, indicating that good medical conditions can relieve the spread of COVID-19 (Li and He, 2020) [19].

Notably, social and economic variables, including population density, had no significant effects on the number of confirmed COVID-19 cases in the target city. This is inconsistent with the views of some researchers who claimed that the high population density of metropolises accelerates the spread of epidemics and that the inhabitants of metropolises are more vulnerable to epidemic outbreaks [24]. The results of the present study revealed that the rationality of the development of metropolitans with high population density shall not be negated due to challenging epidemic prevention and control. If population mobility is adequately controlled, the inhabitants of metropolises may not be vulnerable to epidemic outbreaks. Some scholars conducted a macroscopic, mesoscopic, and microscopic comparison of population density and accumulated number of confirmed COVID-19 cases, with the results suggesting no direct correlation between population density and the spread of COVID-19 [25]. Therefore, in fighting the COVID-19 epidemic, the priority needs to be to minimize population migration, especially from the epicenter, to cut off virus transmission routes, rather than blaming the spread of the epidemic on the high population density. Additionally, urban construction variables, such as road area per capita, domestic garbage per capita, discharged wastewater per capita, and public recreational green space per capita, had no significant effects on the number of confirmed COVID-19 cases in the target city, which is inconsistent with the results of Liu (2020).

### 3.4. Effects of Demographic Characteristics on the Spread of COVID-19

According to the results of the proposed model, population migration from Wuhan and the road distance from Wuhan had significant effects on the spread of COVID-19. However, some cities had relatively low numbers of confirmed COVID-19 cases, despite being located a short distance from Wuhan and having experienced large population migration from Wuhan. Therefore, it is reasonable to deduce that the presence of population characteristics that can hardly be reflected by the proposed model besides population migration, road distance, resident population, and medical conditions, may cause differences in the spread of COVID-19 in cities outside of Hubei, as shown in Figure 7.

First, the numbers of confirmed COVID-19 cases in Wenzhou, Shenzhen, Guangzhou, Shanghai, Beijing, and Chongqing were relatively high, while the population migration from Wuhan was low.

(1) Among the six cities mentioned above, Wenzhou had the second-largest number of confirmed COVID-19 cases, despite its low population migration from Wuhan. On 3 March 2020, the number of confirmed COVID-19 cases in Wenzhou reached 504. This may be due to the fact that the population migration from Wuhan to Wenzhou is dominated by businesspeople. According to statistics from Sohunews, over 170,000 Wenzhou citizens were living in Wuhan in 2017, including businesspeople and their families. For this reason, Wuhan had the second-largest population of Wenzhou citizens (it is even regarded as the “second hometown of Wenzhou citizens”) [26]. During the Spring Festival travel period, various Wenzhou citizens migrated from Wuhan to Wenzhou. Although the absolute size of Wenzhou’s population is not large compared to those of other cities, its businesspeople have a tradition of visiting relatives and friends before the Spring Festival (Xiang and Wang, 2020) [27]. In other words, they had strong mobility in Wenzhou and were, thus, more likely to spread the virus (Shi and Liu, 2020) [21]. Additionally, a large number of businesspeople migrated from other cities to Wenzhou. Indeed, between 24 January and 2 February 2020 (Wenzhou imposed travel restrictions on 2 February 2020), 29,000 people returned to Wenzhou from other cities, an average of about 3000 people per day [28], which increased the difficulty of preventing the spread of COVID-19.

(2) Metropolises, including Beijing, Shanghai, and Guangzhou and Shenzhen, are located in the centers of the Beijing-Tianjin-Hebei, Yangtze River Delta, and Pearl River Delta urban agglomerations, respectively. Compared with Yueyang, Jiujiang, Nanyang, and Zhumadian, which are near Wuhan, these metropolises had similar population migration sizes from Wuhan but larger numbers of confirmed COVID-19 cases. On the one hand, these metropolises have close economic and trade ties with Wuhan, and there is frequent short-term business travel between these metropolises and Wuhan [29]. On the other hand, these metropolises are characterized by large populations, frequent economic activities, and high population mobility. Moreover, their complex public transport networks may accelerate the spread of COVID-19 within the city. As a result, the prevention and control of epidemics in metropolises are extremely challenging. Similarly, some researchers have found that metropolises tend to be associated with severe risks upon sudden public health events [10].

(3) As of 3 March 2020, Chongqing had the maximum number of confirmed COVID-19 cases (576) outside Hubei. Chongqing is characterized by huge labor export (e.g., Chongqing exported 4.74 million laborers to other provinces in 2019) and Hubei is one of the main destinations for this labor (Top 3). As a result, a large number of laborers working in Wuhan returned to Chongqing for family reunions before the Spring Festival in 2020. Additionally, Chongqing is a popular tourism city, and the number of tourists visiting Chongqing during the Spring Festival has been the highest of all cities in China for three consecutive years, resulting in complicated cross-infection during the COVID-19 outbreak; for example, it is possible that tourists from cities outside of Hubei who were SARS-CoV-2 carriers visited Chongqing [30]. Together, these factors contributed to the relatively high number of confirmed cases in Chongqing.

Second, the numbers of confirmed COVID-19 cases in Xinyang, Changsha, Nanyang, and Zhumadian were relatively low. In particular, the number of confirmed COVID-19 cases in Xinyang was not the highest among the four cities, although it had the highest population migration from Wuhan before the Spring Festival. Xinyang, Nanyang, and Zhumadian have frequent personnel exchanges with Wuhan, as they are located near Hubei Province. The population that migrated from Wuhan to these cities before the Spring Festival was dominated by migrant workers. As reported by Luo et al. (2017), the collective influence of individuals on the social network is highly correlated with their economic status [31]. In other words, since migrant workers have low economic status and the frequency of their social activity is significantly lower than businesspeople with high economic status, it is likely that the low local mobility of migrant workers decelerated the spread of COVID-19.

Notably, cities in Henan Province immediately took strong protective measures during the COVID-19 outbreak. At the end of 2019, shuttle buses between these cities and Wuhan were suspended and the public was reminded to pay attention to the latest situation of the spread of COVID-19 via live television and software messages [32]. Meanwhile, relevant measures were executed even at the grass-roots level in villages and towns. For instance, all citizens in Chaigou Village of Gongyi City were motivated to isolate villagers returning from Wuhan for a 14-day observation period. The villagers were repeatedly reminded about COVID-19 by announcements, banners, and village horns [33]. Additionally, Caizhuang Town of Kaifeng City required all villages to close roads to reject outsiders [34]. These measures had a positive effect on the prevention and control of COVID-19 (Chen et al., 2020) [35]. Tian et al. (2020) also confirmed that some emergency response measures, including suspending public transportation and closing entertainment venues, were related to the reduction of the epidemic [36].

Thus far, we have discussed the relationship between the population migration from Wuhan and the number of confirmed COVID-19 cases in each city based on linear regressions and scatter plots. Our conclusions may have several limitations. First, this study only considered population migration from Wuhan, while population migration from other key epidemic regions and inter-city population migration were not included. Secondly, the possible stationary relationship between the dependent and independent variables in the linear regression may affect the accuracy of the results. Further analysis of migration big data can expand the depth and scope of analysis to achieve thorough investigations of population migration dynamics during major public health events such as the COVID-19 epidemic. In the future, with increased data availability, more appropriate methods can be used to test the relationship between the number of confirmed cases and population migration.

## 4. Conclusions

This study demonstrated that population migration from the outbreak center was the most important factor affecting the spread of COVID-19 in different Chinese cities. Therefore, in future disease outbreaks, the outbreak center should be identified as soon as possible and measures should be taken immediately to minimize population migration from this center. Meanwhile, people who come into contact with people who have migrated from the outbreak center should be screened immediately to cut off the infection route.

Additionally, the spread of COVID-19 in cities outside of Hubei was affected by demographic characteristics of population migration from Wuhan, besides total population migration, road distance from Wuhan, GDP per capita, resident population, and medical conditions. For instance, the numbers of confirmed COVID-19 cases in cities whose population that migrated from Wuhan was dominated by businesspeople tended to be higher than in cities whose population migrating from Wuhan was dominated by migrant workers, since the social activity frequency of businesspeople is generally higher than that of migrant workers. Therefore, for the prevention and control of the COVID-19 epidemic, more attention should be paid to groups that perform frequent social activities, such as the merchants at the epidemic epicenter, and measures should be taken to reduce the possibility of their spreading the virus. Meanwhile, the spread of COVID-19 in cities was also found to be affected by other factors, including economic ties with Wuhan, the characteristics of urban development, and specific prevention measures.

A review of the spread of COVID-19 in China is essential to provide references in social and economic aspects for the prevention and control of major public health events in other countries and in the future. China’s experiences in the prevention and control of COVID-19 suggest that effective traffic restriction and isolation measures must be taken and specific measures should be designed according to the characteristics of population migration at the early stage of a major public health event.

## Figures and Tables

**Figure 1 ijerph-18-03255-f001:**
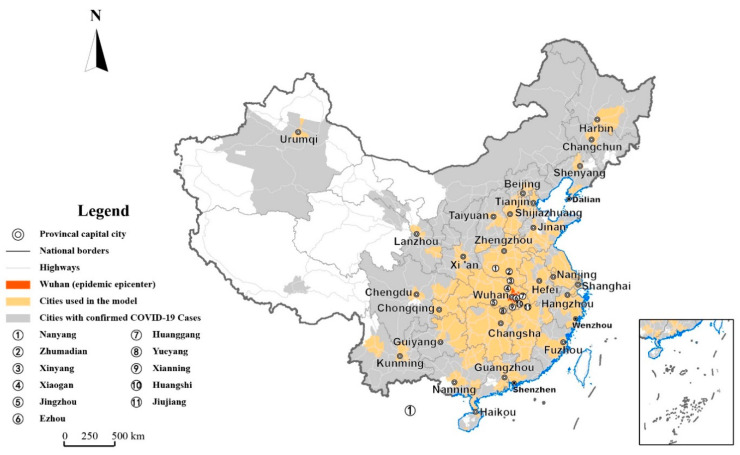
The geographic location of the cities involved in this study.

**Figure 2 ijerph-18-03255-f002:**
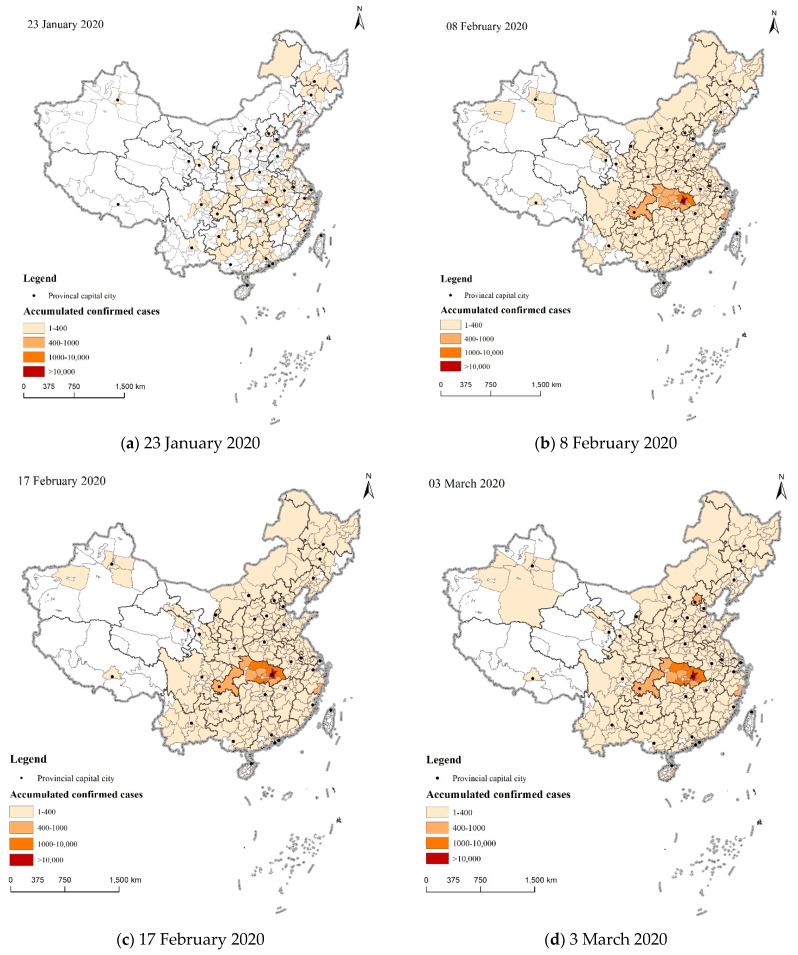
Accumulated numbers of confirmed COVID-19 cases in China on different dates. (**a**) 23 January 2020; (**b**) 8 February 2020; (**c**) 17 February 2020; (**d**) 3 March 2020.

**Figure 3 ijerph-18-03255-f003:**
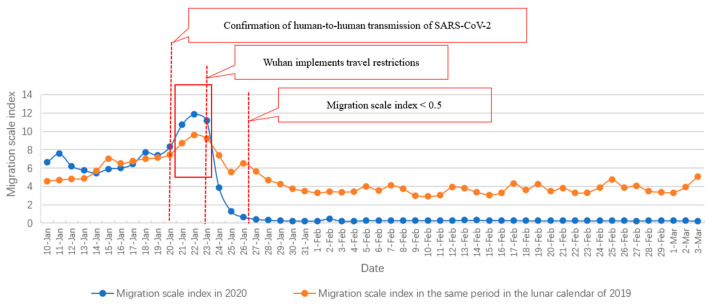
The trend of population migration from Wuhan during the Spring Festival in 2019 and 2020.

**Figure 4 ijerph-18-03255-f004:**
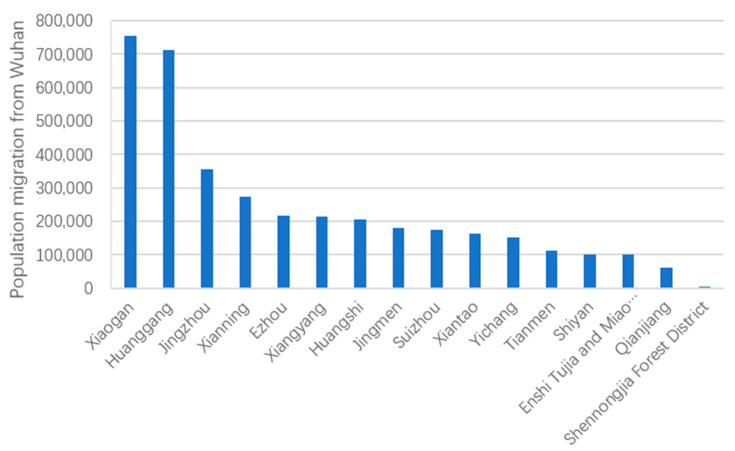
Population migration from Wuhan to cities in Hubei Province.

**Figure 5 ijerph-18-03255-f005:**
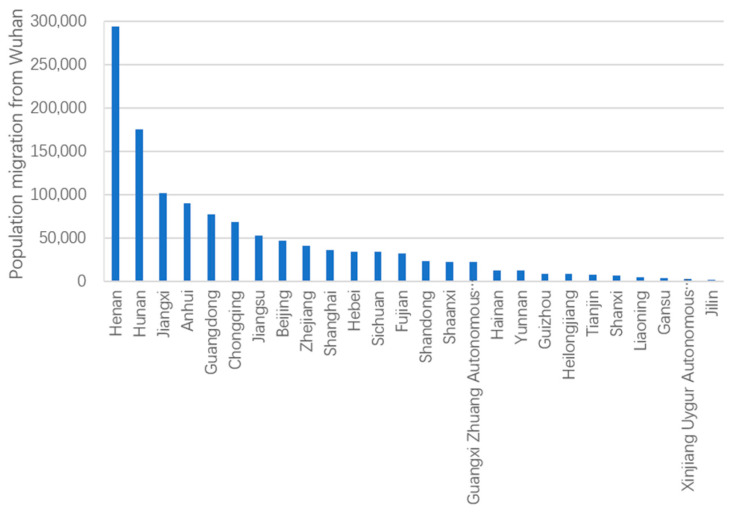
Population migration from Wuhan to other provinces.

**Figure 6 ijerph-18-03255-f006:**
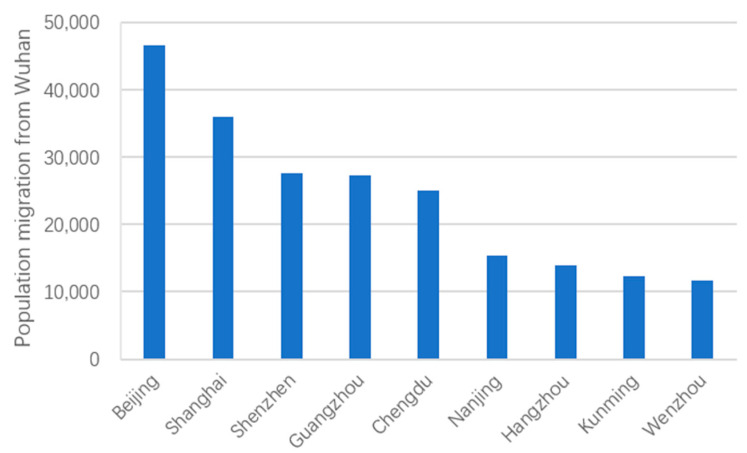
Population migration from Wuhan to other key cities.

**Figure 7 ijerph-18-03255-f007:**
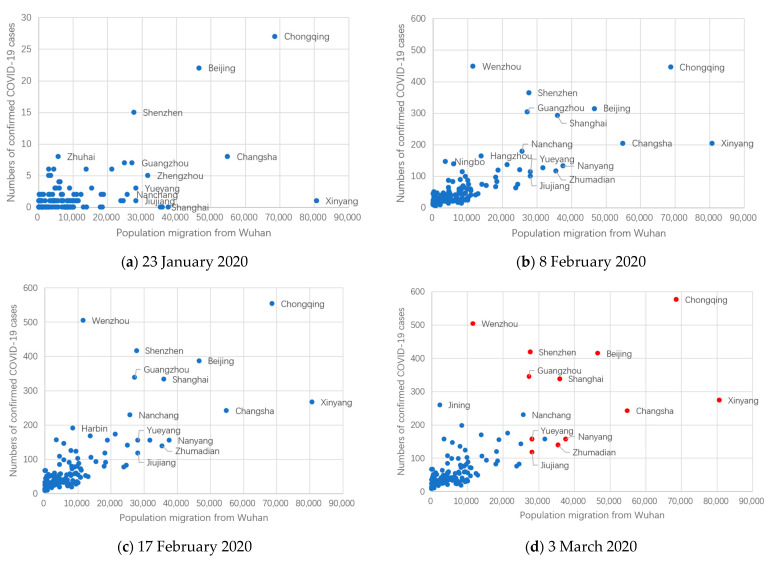
Numbers of confirmed COVID-19 cases and population migration from Wuhan for cities outside of Hubei. The *x*-axis shows the estimated population migration from Wuhan to different cities and the *y*-axis shows the accumulated number of confirmed COVID-19 cases in different cities. (**a**) 23 January 2020; (**b**) 8 February 2020; (**c**) 17 February 2020; (**d**) 3 March 2020.The red dots in (**d**) show the key cities that were focused on in this section.

**Table 1 ijerph-18-03255-t001:** Definitions of variables.

Variable	Definition	Unit
Disease	Accumulated number of confirmed COVID-19 cases	Persons
Migration	Population migration from Wuhan between 10 and 26 January 2020	Persons
Dist	Road distance from Wuhan	Kilometers
Pergdp	GDP per capita	Yuan
Population	Resident population at the end of 2018	10,000 persons
Hospital	Number of hospital beds per 1000 residents	-
Pop_density	Population density	Persons/km^2^
Road	Road area per capita	m^2^
Garbage	Domestic garbage collected and transported per capita	kg
Wastewater	Annual quantity of wastewater per capita	m^3^
Greenspace	Public recreational green space per capita	m^2^
Capital_city	Is the city the capital of the province? (Yes = 1, No = 0)	N.A.

**Table 2 ijerph-18-03255-t002:** Correlations between explanatory variables.

Variable	(1)	(2)	(3)	(4)	(5)	(6)	(7)	(8)	(9)	(10)
(1) Migration	1.000									
(2) Dist	−0.591	1.000								
(3) Pergdp	0.070	0.223	1.000							
(4) Population	0.154	0.207	0.091	1.000						
(5) Hospital	0.058	0.283	0.503	0.212	1.000					
(6) Pop_density	0.112	−0.011	−0.268	0.271	−0.036	1.000				
(7) Road	−0.121	−0.204	−0.001	−0.180	−0.254	−0.242	1.000			
(8) Garbage	0.012	0.204	0.307	−0.081	0.038	−0.258	0.190	1.000		
(9) Wastewater	−0.097	0.196	0.425	−0.005	0.130	−0.148	0.153	0.359	1.000	
(10) Greenspace	−0.251	0.184	0.240	0.044	−0.131	−0.278	0.328	0.157	0.175	1.000

**Table 3 ijerph-18-03255-t003:** Empirical estimation results with dependent variables of the accumulated number of confirmed COVID-19 cases in cities on 23 Jan, 8 Feb, 17 Feb, and 3 Mar.

Variable	Full Sample				Subsample without Hubei Province			
	(1)	(2)	(3)	(4)	(5)	(6)	(7)	(8)
	23 Jan	8 Feb	17 Feb	3 Mar	23 Jan	8 Feb	17 Feb	3 Mar
Migration	−0.071 **	0.082 ***	0.079 ***	0.080 ***	−0.072 **	0.060 ***	0.058 ***	0.058 ***
	(−2.22)	(5.41)	(5.20)	(5.23)	(−2.26)	(4.16)	(3.98)	(4.00)
Dist	0.015	−0.115 ***	−0.117 ***	−0.114 ***	−0.057	−0.070 *	−0.067 *	−0.064
	(0.17)	(−3.15)	(−3.08)	(−2.98)	(−0.62)	(−1.79)	(−1.66)	(−1.55)
Pergdp	−0.086	0.124 ***	0.119 ***	0.115 ***	−0.049	0.119 ***	0.112 ***	0.109 ***
	(−0.77)	(3.28)	(3.32)	(3.19)	(−0.46)	(3.14)	(3.15)	(3.03)
Population	−0.020	0.096 ***	0.086 ***	0.090 ***	−0.020	0.132 ***	0.124 ***	0.128 ***
	(−0.27)	(4.16)	(3.83)	(3.88)	(−0.29)	(5.06)	(4.90)	(4.94)
Hospital	0.074	−0.059 **	−0.044	−0.041	0.018	−0.074 ***	−0.059 **	−0.055 **
	(0.62)	(−2.04)	(−1.65)	(−1.49)	(0.16)	(−2.65)	(−2.27)	(−2.10)
Pop_density	−0.005	−0.008	−0.009	−0.015	0.035	0.012	0.010	0.004
	(−0.06)	(−0.27)	(−0.35)	(−0.57)	(0.44)	(0.43)	(0.37)	(0.15)
Road	0.103	−0.009	−0.016	−0.007	0.137	0.001	−0.005	0.005
	(1.03)	(−0.26)	(−0.50)	(−0.20)	(1.31)	(0.02)	(−0.16)	(0.16)
Garbage	−0.045	0.018	0.031	0.022	−0.004	0.065	0.074	0.065
	(−0.25)	(0.26)	(0.49)	(0.34)	(−0.02)	(0.92)	(1.14)	(0.97)
Wastewater	−0.250	0.046	0.024	0.020	−0.371 ***	0.046	0.027	0.023
	(−1.64)	(0.97)	(0.52)	(0.43)	(−2.67)	(1.01)	(0.60)	(0.51)
Greenspace	−0.029	0.010	0.031	0.041	−0.094	−0.001	0.018	0.029
	(−0.13)	(0.12)	(0.45)	(0.60)	(−0.42)	(−0.01)	(0.26)	(0.41)
Capital_city	−0.143	−0.003	−0.006	−0.007	−0.109	0.017	0.012	0.011
	(−1.10)	(−0.08)	(−0.16)	(−0.18)	(−0.83)	(0.39)	(0.30)	(0.26)
_cons	3.670 **	−0.930 *	−0.689	−0.619	3.737 **	−1.693 ***	−1.465 ***	−1.402 ***
	(2.57)	(−1.75)	(−1.45)	(−1.29)	(2.56)	(−3.10)	(−2.95)	(−2.83)
N	79	131	131	131	73	120	120	120
R2	0.260	0.708	0.719	0.708	0.305	0.621	0.632	0.619

Notes: Columns 1–4 show the regression results using all the samples, and columns 5–8 show the regression results using the subsample without Hubei Province. Heteroscedasticity-robust *t* statistics are shown in parentheses; * *p* < 0.10, ** *p* < 0.05, *** *p* < 0.01.

## Data Availability

Publicly available datasets were analyzed in this study. Data of confirmed COVID-19 cases can be collected from the websites of health commissions of prefecture-level cities in China. Big data of migration can be found here: https://qianxi.baidu.com (accessed on 1 August 2020). The data of road distances can be found here: https://map.baidu.com (accessed on 1 August 2020). Other data can be found here: https://data.cnki.net/ (accessed on 12 August 2020).

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
