# Peer review of "What Are the Reasons for the Different COVID-19 Situations in Different Cities of China? A Study from the Perspective of Population Migration"

_ijerph, 2021, doi:10.3390/ijerph18063255_

Round 1

Reviewer 1 Report

The Yanting et al. Manuscript is a highly valuable topic related to the prevention and control of an epidemic outbreak such as that caused by COVI-19. In general, it is well written, and its contribution is very valuable in relation to the comparisons for the control of population outflow control. However, I have three points that the authors should review and indicate about their statistical analyzes.

  1. Their regression model fits could be compromised because lines 171-180 do not indicate whether they verified the assumptions of the statistical technique. A condition that is important to show as evidence, since the requirement is to comply with the normal error distribution and homogeneity of variances. Failure to comply with these assumptions there are procedures that are applied to solve this limitation. Otherwise, a type II statistical error is incurred. It is important that the authors carry out the review since Table 2 (line 348) indicates the values of the t-tests associated with the analysis and this technique is subject to fulfilling the assumptions indicated above. In addition, in the header of Table 2, you must indicate what the values indicated in parentheses (1), (2) ... (8). So that the reader can clearly identify its meaning.
  2. Figures 2-5 do not have a line on the y-axis (figure 6 if the axis line is defined) homogenize design in the graphs. In figure 2, the legend corresponding to the y axis must be indicated. Also, in Figure 3-4, change outflow to Outflow (initial letter capitalized).
  3. In figure 6, the projection of the dotted line is not indicated what it means, is it a linear regression fit? If so, I suggest locating the regression equation in its corresponding graph. As seen in the bivariate dispersion, it is quite possible that the assumption of normal error distribution and homogeneity of variances was not met. Review and indicate it in the corresponding section.

Reviewer 2 Report

The authors analyzed the differences in the spread of COVID-19 in different cities of China from the perspective of population migration. It is good work, but I have some major concerns on the methodology:

What was the cross-correlation between independent variables? Was it even checked before performing the analysis? Some of the high R2 values in Table 2 may be the result of the redundant variables. Provide a table with cross-correlations.

Was the biasness of the model checked? Authors should perform Jarque-Bera (or similar) diagnostics and report the result in the paper.

Few minor concerns/suggestions:

Authors should add a map (as Figure 1) showing the Wuhan and other main cities (mentioned in the manuscript) as labels. It may be good to add highways as well.

How distance from Wuhan to other cities was calculated? I know the authors used Baidu digital map but were it by Euclidean distance or Road distance?

In the Introduction (Line 29-31), instead of 20 July, the latest numbers should be mentioned.

Line 83-84: What is Cordon Sanitaire?

Figure 1: What classification was used in these maps? It seems manual. How the classes 1-50, 51-300, and so on were chosen? Do we really need all 5 classes? Zero may be removed since it is obvious, and four classes could be better. Further, 400 cut off (Line 217) was used; it should match with the class breaks. Please also change the scale from 1,740 to either 1,500 or 2,000.

Line 343: I really don’t know how population density can SIMPLY be reduced?

What are the red dots in Figure 6?

The conclusion contains text that should be part of the Introduction or discussion. For instance, the text from Line 440 to 445 should be in the Introduction. Limitation (the last paragraph) should be in the discussion. Add other limitations from the method (linear regression) as well including collinearity, the stationary relationship between the dependent and independent variables, etc.

Perform a good review to remove to edit the paper. What is “canter” on Line 320? I think it should be the center. Similarly, replace “?” on line 180 with the random error symbol.

Round 2

Reviewer 2 Report

Thank you for addressing the comments.  There are still few issues:

  1. The map in Figure 1 is not clear. It is not a high-quality good map. Labels are not readable. Use the halo effect or move them around with leader lines.
  2. Euclidean distance: How is it relevant? Why not actual road distance that depicts the way people travel in the real world? Is it too much work or was it not possible to calculate it? Also, was the center of the cities was used to calculate the distances? Was it a geographic center or a population-weighted center? Why? In most cases, Euclidean distance and road distances are highly correlated and either could be used in analysis/modeling with proper justification. Please refer to the following manuscript: https://bmchealthservres.biomedcentral.com/articles/10.1186/1472-6963-9-200

Besides these comments, I do not have any problem with the manuscript proceeding to publication
